# The Influence of Adding B_4_C and CeO_2_ on the Mechanical Properties of Laser Cladding Nickel-Based Coatings on the Surface of TC4 Titanium Alloy

**DOI:** 10.3390/ma17153823

**Published:** 2024-08-02

**Authors:** Shanna Xu, Keqi Han, Haili Wang, Yuntao Xi, Lei Wang, Xikai Dong

**Affiliations:** 1Xi’an Key Laboratory of High Performance Oil and Gas Field Materials, School of Material Science and Engineering, Xi’an Shiyou University, Xi’an 710065, Chinaxikaidong41@gmail.com (X.D.); 2Xi’an Rare Metal Materials Institute Co., Ltd., Xi’an 710016, China

**Keywords:** laser cladding, TC4 titanium alloy, nickel-based composite coating, microhardness

## Abstract

The development of titanium alloys is limited by issues such as low hardness, poor wear resistance, and sensitivity to adhesive wear. Using laser cladding technology to create high-hardness wear-resistant coatings on the surface of titanium alloys is an economical and efficient method that can enhance their surface hardness and wear resistance. This paper presents the preparation of two types of nickel-based composite coatings, Ni60-Ti-Cu-xB_4_C and Ni60-Ti-Cu-B_4_C-xCeO_2_, on the surface of TC4 titanium alloy using laser cladding. When the B_4_C addition was 8 wt.%, the hardness of the cladding layer was the highest, with an average microhardness of 1078 HV, which was 3.37 times that of the TC4 substrate. The friction coefficient was reduced by 24.7% compared to the TC4 substrate, and the wear volume was only 2.7% of that of the substrate material. When the CeO_2_ content was 3 wt.%, the hardness of the cladding layer was the highest, with an average microhardness of 1105 HV, which was 3.45 times that of the TC4 substrate. The friction coefficient was reduced by 33.7% compared to the substrate material, and the wear volume was only 1.8% of that of the substrate material.

## 1. Introduction

Titanium alloys have become a research and application direction for the next generation of high-performance petroleum pipe materials due to their excellent properties that cannot be compared to those of steel materials, such as high strength, low density, excellent corrosion resistance, good high-temperature resistance, high fatigue resistance, low elastic modulus, and non-magnetism [1]. The wear of titanium alloy drill pipes under working conditions is still not as good as that of steel drill pipes, and it is important to improve the wear resistance of titanium alloys [2]. Wear-resistant belts of drill pipe joints have been widely applied in deep well drilling, large-displacement well drilling, and high-angle well drilling engineering. However, the gas metal arc welding method used for steel drill pipes is not suitable for titanium alloy drill pipe surfaces [3]. It is urgent to improve the wear resistance of titanium alloy drill rods through advanced surface modification technologies for titanium alloys. Surface modification technologies for titanium alloys include electroplating, electroless plating, thermal spraying, nitriding and carburizing, micro-arc oxidation, and laser cladding [4]. Coatings prepared by laser cladding technology have the advantages of simple operation, high production efficiency, easy automation, controllable coating thickness, good bonding with the matrix, etc. [5].

The cladding powder material is the key factor affecting the forming and comprehensive properties of the cladding layer. The cladding materials that can improve the hardness and wear resistance of titanium alloys mainly include Ni, B, C, Cr_2_C_3_, TiC, WC, TiB, TiB_2_, etc. [6,7,8,9,10]. Ni60 nickel-based alloy has the characteristics of good comprehensive performance, good wear resistance, and corrosion resistance, and it is a common laser cladding material for preparing wear-resistant coatings [11,12,13]. It has strong deoxidation and self-melting effects, but in the laser cladding production process, if the process parameters are not properly set, the cladding layer is prone to cracks and pores [14]. Zhang [15] found through research that nickel-based alloy coating is helpful to improving the surface properties of TC4 titanium alloy, but the mismatch of thermal physical properties, such as thermal expansion coefficients, may lead to cracks caused by residual thermal stress. Yao et al. [16] found that Ti_2_Cu intermetallic compounds mainly exist in spherical and acicular forms, among which the acicular strengthening effect is more obvious. Adding the Cu element to nickel-based coatings can improve the corrosion resistance of the coating by increasing the density of the passivated film [17]. Yang et al. [18] analyzed the effect of Cu content on the microstructure and properties of Ti-Cu-N coating. The results showed that the coating was mainly composed of TiN, CuTi_3_, and a small amount of TiO_6_. With the increase in Cu content, the coating hardness decreased, and the friction coefficient increased, but it was still lower than that of the matrix, and CuTi_3_ and TiN improved the wear resistance of the composite coating. The Cu element has the effect of refining the grain, and during cooling and solidification, it will form a solid solution uniformly existing in the cladding layer. Osorio et al. [19] proposed that copper can promote a uniform and smooth oxidation film of Ti-Cu. Therefore, in this paper, Ti and Cu elements are added to the cladding material of a nickel-based alloy in order to solve the problem of easy cracking of nickel-based alloy coatings and improve the hardness and wear resistance of the alloy coating.

Ceramic materials have high hardness, light weight, excellent wear resistance, and high temperature performance. Ceramic materials are combined with metal alloy coatings to prepare ceramic-metal composite coatings, which have the high hardness and wear resistance characteristics of ceramic materials and high toughness of metal materials [20]. There are two ways to prepare the ceramic reinforcement phase in composite coatings by laser cladding: in situ synthesis and direct addition. Direct addition of the ceramic phase leads to lower toughness and high crack sensitivity of the coating, while the ceramic reinforcement phase generated in situ by designing the cladding material composition has strong bonding with the matrix, which can improve the hardness and wear resistance of metal-based coatings [21]. B_4_C material has the lowest density of ceramic materials, only 2.52 g/cm^3^, and its titanium alloy can be synthesized in situ through metallurgical reaction in the process of laser cladding to form the ceramic reinforcement phase [22]. The thermal expansion coefficients and densities of TiB and TiC are close to those of titanium alloy, which is the best reinforcement phase for preparing wear-resistant coatings on titanium alloy surfaces [23]. When preparing wear-resistant coatings, rare earth oxides such as CeO_2_, Y_2_O_3_, and La_2_O_3_ are added to cladding materials, which can not only refine the grain but also purify the molten pool, promote the flow of the molten pool, and reduce the porosity of the cladding layer [24]. Therefore, in order to improve the hardness and wear resistance of Ni60 alloy coating, Ni-base composite coating was prepared by adding Ti, Cu, B_4_C, and CeO_2_ to Ni60 alloy powder for in situ synthesis of the ceramic reinforcement phase, and the effects of the B_4_C and CeO_2_ contents on the microstructure and properties of the composite coating were investigated.

This article utilizes laser cladding technology, using Ni60, Cu, Ni, B_4_C, and CeO_2_ as alloy powders, to create composite coatings on a TC4 substrate. The microstructure and properties of the coatings were tested to provide a reference for repairing or enhancing the surface properties of drill rod joints and extending their service life.

## 2. Materials and Methods

The substrate selected in this experiment was a TC4 (Ti-6Al-4V) titanium alloy rolled sheet, shaped 100 mm × 100 mm × 10 mm, with moderate strength. The chemical composition is shown in Table 1. The microstructure of the TC4 substrate material is shown in Figure 1. The α and β phases can be clearly observed in the figure. Since the base material is a TC4 rolled sheet (Western Metal Materials Co., Ltd., Xi’an, China), it can be seen that the α and β phases are fibrous along the rolling direction.

In this paper, the laser cladding process parameters for preparing Ni60 nickel-based alloy coating were optimized. Ni60 alloy contains two elements, Si and B, which easily react with oxygen to form borosilicate when melting and has the effect of deoxidation and slag-making [25]. The cladding layers prepared by laser cladding technology using Ni60 alloy powder have good toughness, corrosion resistance, and wear resistance, so it is widely used in practical production. Table 2 shows the chemical composition of Ni60 alloy powder with Ti and Cu powders. On the basis of adding Ti and Cu to Ni60 nickel-based alloy powder, different contents of B_4_C were added, and the hard reinforcement phases, such as TiB, TiC, and TiB_2_, were expected to be synthesized in situ. After selecting the best ratio of cladding powder, different contents of CeO_2_ were added to explore the influence of CeO_2_ on the forming quality, phase composition, microstructure, microhardness, wear resistance, and corrosion resistance of the cladding layer. The micromorphologies of different kinds of cladding powders and several mixed cladding powders are shown in Figure 2. The powder size of Ni60 is ~44 μm and the purity is 99.5%. Ti, Cu, B_4_C, and CeO_2_ are all ultrafine powders with a purity of 99.9%.

The main equipment in this experiment included a IHN-1GX-3000P fiber laser with an output power of 3 kW, a carrier gas powder feeding system with a capacity of 10 L, a water cooler, and a high negative-pressure dust purifier. The powder feeding mode was blown powder feeding in different axes. In the process of laser cladding, high purity argon gas was used as the shielding gas and for powder transportation. The powder was dried in an XGQ-2100 electric thermostatic drying oven (YiSuDa Co., Ltd., Suqian, China) at 75 °C for 2 h. The powder was stirred mechanically for 3 h with a powder mixer to fully mix the metal powder. Before cladding, the oxide layer on the titanium alloy surface needed to be ground with sandpaper, and then the surface was cleaned with anhydrous ethanol and dried. Reasonable output energy is the key to the preparation of a better coating, and the laser power and scanning speed are the two most important process parameters that affect the output energy. In the early stage of this experiment, the process parameters were optimized. When the scanning speed was 4 mm/s, the microhardness of the cladding layer had the least fluctuation, in the range of 800~1200 W power. The distribution along the cladding layer was uniform, which indicated that the scanning speed reasonably matched the laser power. The surface of the sample was polished with 180~3000 mesh water-abraded sandpaper. Finally, the cross-section of the cladding layer was mechanically polished with 5.0 μm and 1.0 μm diameter alumina polishing agent until the surface of the sample was specular. An etching agent with a ratio of HF/HNO_3_/H_2_O = 2:3:5 (volume ratio) was used to etch the metallographic samples for 10 s, and then the samples were washed and dried with a large amount of anhydrous ethanol after etching. A Neophot-21 metallographic microscope (OM) was used to observe the cross-section morphology and microstructure of the cladding layer. A NovaNano 450 scanning electron microscope (SEM) was used to observe the cross-sectional microstructure, surface friction, and wear morphology of the samples, and energy dispersion spectrometer (EDS) was used to analyze the element distribution of each phase of the cladding layer with different compositions. The phase composition of the cladding layers with different compositions was determined by X-ray diffraction (XRD) using Shimadzu model XRD-6000 (Kyoto, Japan). In order to study the variation in microhardness from the top center of the cladding layer to the TC4 matrix material, an HRD-1000TMC/LCD (Kexin Co., Ltd., Suzhou, China) turret digital microhardness tester was used for measurement. The load was 1 kg and the load retention time was 15 s. In order to ensure the stability of the data, three sets of measuring points were arranged, and the interval between each group was 0.1 mm. A multi-functional friction and wear testing machine (MMX-3G, Jinan HengXu Testing Machine Technology Co., Ltd., Jinan, China) was used for friction and wear experiments. The friction pair was a pin plate made of GCr15 bearing steel with an inner diameter of 38 mm and an outer diameter of 54 mm. The friction pin was the TC4 sample coated with different components as the wear surface of the friction and wear test. The size of the friction pin sample was a cylinder with a diameter of 4.8 mm.

## 3. Results and Discussion

### 3.1. The Effect of B_4_C Content

With the development of laser cladding technology, the selection of cladding materials has changed from single alloy coatings to ceramic-metal composite coatings. When the bonding strength between the ceramic material and the substrate is low, and the thermal expansion coefficient is different from that of the titanium alloy substrate, the coating easily cracks and falls off, so ceramic coatings are rarely prepared directly on the surfaces of titanium alloys. Ceramic-metal composite coatings synthesize the excellent properties of ceramics and metals and improve the strength and hardness of alloy coatings. By adding B_4_C particles to TiNi-Ti_2_Ni cladding powder, Feng et al. generated TiB and TiC hard ceramic phases in situ in the cladding layer [26]. The addition of B not only promoted the formation of the hard phase of TiB_2_ but also had a solid solution-strengthening effect on the cladding layer due to the small atomic radius of the B element and the large difference between the atomic radii of other metal elements, thus improving the hardness of the cladding layer [27]. Adding B_4_C powder can refine the grain when strengthening a nickel-based coating with B_4_C, but excessive B_4_C will make the fluidity of the molten pool deteriorate, resulting in coating defects [28]. Therefore, it is of great significance to study the addition of B_4_C in the Ni60 + Cu + Ti cladding material system.

#### 3.1.1. Microstructure of the Cladding by B_4_C Adding

The cladding material used was a mixture of Ni60, Ti, Cu, and B_4_C powders, in which the mass ratio of Ni60/Ti/Cu was 60:25:15, and the content of B_4_C was variable. The specific compositions of the cladding powders are shown in Table 3.

Figure 3 shows the cross-section morphologies of cladding layers with different B_4_C contents under an optical microscope, which can be roughly divided into four regions: the cladding layer (CZ), fusion line (FL), the heat-affected zone (HAZ), and the bulk material (BM). The following conclusions can be drawn from the figure: the cladding layers with different B_4_C contents are well combined with the substrate. The cladding layer with 10 wt.% B_4_C content had obvious cracks, while the cladding layers with other B_4_C contents had dense and uniform structures and no obvious cracks, pores, and other defects. There are two main reasons for the cracks in the cladding layer: with the addition of B_4_C powder, there is a large residual stress in the cladding layer after the cooling process of the melt pool due to the difference in the linear expansion coefficients of each material. The excessive addition of B_4_C powder makes the fluidity of the molten pool deteriorate during the cladding process, resulting in the formation of cracks in the metal molten liquid during solidification. Interatomic mixing occurs between the cladding layer melt and the matrix melt in the bonding zone during the solidification process, which is metallurgical bonding with high bonding strength.

The X-ray diffraction patterns of the cladding layers with B_4_C mass fractions of 4% and 8% are shown in Figure 4a. The results showed that no B_4_C was found in the coatings with different compositions, indicating that the B_4_C powder had been completely reacted. The cladding layers with different B_4_C contents contained basically the same main phase. In the laser cladding process, the cladding powder and the surface layer of the substrate material absorbed heat, melted to form a molten pool, and upon cooling, the ceramic reinforcement phases were TiC and TiB. In the process of preparing a Ni60-Ti-Cu-xB_4_C nickel-based composite coating by laser cladding on the surface of a TC4 alloy, a metallurgical chemical reaction will occur between the surface of the TC4 alloy and the molten cladding material. Due to the addition of multiple elements, the chemical reactions occurring in the melt pool are very complicated. Thermodynamic calculation software (HSC Chemistry6.0) was used to calculate the standard Gibbs free energy of the five chemical reactions, as shown in Table 4. The Gibbs free energy in thermodynamics can be used to determine whether chemical reaction equations can occur and can also be used to determine the spontaneous tendency of each reaction. Figure 4b shows the above reactions at different reaction temperatures. It can be seen from the figure that the standard Gibbs free energy of the five intermediate binary compounds generated by metallurgical chemical reactions was <0 in the temperature range of 300~2000 K, all of which were exothermic reactions, indicating that the chemical reactions listed in Table 4 could be carried out spontaneously and thermodynamically. The absolute values of TiC and TiB were the largest, which indicated that the reactions tended to occur the most in thermodynamic terms. In the process of laser cladding, the chemical reactions to form TiC and TiB preferentially occurred, and the heat released by the reaction could promote other chemical reactions. Figure 4c shows the change curve of the product at different temperatures. The stability of the product was sorted (<500 k) from strongest to weakest, as follows: TiB_2_, TiC, TiB, NiTi_2_, NiTi, and Ti_2_Cu. Since the chemical reactions to form TiC and TiB occurred first, although TiB_2_ was more stable than TiB, laser cladding is a rapid solidification process, so only a small part of TiB combined with the free B atoms in the molten pool to form TiB_2_. Since NiTi_2_ produced by chemical reactions was more stable than NiTi, it was preferentially produced. TiC, TiB, and NiTi_2_ should be the main products.

Laser cladding for coating preparation is completed in a very short time; therefore, the constituents of the solid solution cannot mix sufficiently, resulting in a non-equilibrium solidification process. As a result, various microstructures exist within the cladding layer. The macroscopic microstructural morphology of the cladding layers with 8% B_4_C content is shown in Figure 5a. The bonding zone serves as a transition between the TC4 substrate and the cladding layer, and its defect-free condition determines the quality of the bond between the cladding layer and the substrate material. Observations from Figure 5a revealed that there were no microcracks, pores, or inclusions in the bonding zone. The fusion line was intact, clear, and smooth. Atomic mixing occurred between the molten pool of the cladding layer and the substrate during the solidification process, resulting in a high-strength metallurgical bond. Below the fusion line on the side of the substrate material was the heat-affected zone, which was not easily corroded by etching solutions, and thus its microstructural morphology was not apparent. Above the fusion line was the region of the cladding layer, the microstructural morphology of which was mainly related to the ratio (G/R) of the temperature gradient (G) and solidification rate (R) in the liquid phase at the solid–liquid interface front, as seen in Figure 5b. At the bottom of the cladding layer, near the substrate material, the temperature gradient (G) was very large. Here, the G/R ratio tended toward infinity, meaning that nucleation occurred faster than crystal growth in the liquid phase. Therefore, the liquid metal near the substrate material cooled down and formed planar crystals growing with nucleation cores as the interface. During the solidification process, the latent heat of crystallization is released. As the solid–liquid interface continued to move into the liquid phase, the temperature gradient (G) decreased, and the solidification speed (V) relatively increased. Consequently, the G/R ratio also decreased. The temperature dropped fastest perpendicular to the fusion line, and the liquid metal ahead of the solid metal front tended to form columnar crystals oriented vertically to the fusion line. As G/R decreased, the grain size was gradually reduced, transitioning from columnar crystals to dendrites, and then to fine equiaxed grains [29].

#### 3.1.2. Microhardness

Figure 6 shows the distribution curve of the microhardness of cladding layers with different B_4_C contents from the surface to the substrate. It can be seen from the figure that the distribution law of the hardness of cladding layers with different B_4_C contents was similar in the direction from the top of the cladding layer to the substrate, and the microhardness at the top of the cladding layer was the highest due to the dilution effect of the substrate material on the coating.

The reinforcement phase near the bottom decreased and the grain size was coarse, which made the hardness value near the bottom of the cladding layer low, and the hardness value dropped sharply to the hardness of the substrate material at the interface. With the increase in the amount of B_4_C added, the hardness value of the cladding layer showed an overall increasing trend. Among them, the coating with 8% B_4_C content had the highest hardness, and the hardness value was up to 1377 HV_1_, which was 4.05 times higher than that of the TC4 substrate material. With the increase in B_4_C addition, the number of in situ-generated reinforcing phases increased, which had a certain effect on enhancing the average hardness values of the cladding layers. During the cladding process, B_4_C absorbed laser energy to release free B and C elements, which directly formed ceramic reinforcing phases such as TiC and TiB with the Ti element. The cladding layer containing 8% B_4_C had the highest quantity of hard reinforcing phases and a more uniform distribution, resulting in the highest average hardness value.

#### 3.1.3. Friction and Wear Property

It can be seen from Figure 6 that the cladding layer containing 0% B_4_C had the lowest hardness value, while the cladding layer containing 8% B_4_C had the highest hardness value. Under the same wear conditions, the wear resistance of the material is correlated with the hardness of the material. Therefore, the coatings containing 0% B_4_C and 8% B_4_C were selected for friction and wear performance testing. Figure 7a shows the friction coefficient curves of the TC4 substrate material and the 0% B_4_C and 8% B_4_C cladding layers. At the initial stage of the test, the specimen pin and the auxiliary friction disc began to be in linear contact. With the increase in wear time, the contact area increased, and the friction coefficient rose rapidly and then decreased slightly, and all entered the stable friction interval for about 400 s, because the wear area did not change after a certain period of time, and the friction coefficient tended to be stable. The average friction coefficients of the TC4 substrate material and the 0% B_4_C and 8% B_4_C laser cladding layers were 0.3725, 0.3301, and 0.2804, respectively, which were reduced by 11.4% and 24.7% compared to the TC4 substrate material. It can be seen that the friction coefficient of the cladding layers with different compositions was significantly lower than that of the TC4 titanium alloy substrate material, which had a certain degree of improvement on the surface of the substrate material. The size of the friction coefficient is mainly determined by the state and quality of the friction surface. Large friction surface stiffness and low surface roughness can effectively reduce the friction coefficient. Therefore, in the process of friction and wear, the two cladding layers containing 0% B_4_C and 8% B_4_C had greater stiffness and lower roughness than the friction surface of TC4. TiB and TiC had the characteristics of high hardness and low friction coefficients, among which TiB had a stronger ability to resist the influence of dry friction and wear and the ability of coating spalling. The nickel-based composite coating containing 8% B_4_C had the lowest friction coefficient, because the number of TiB and TiC phases generated in the original location of the cladding layer was the largest, and the stiffness was the largest.

As shown in Figure 7b, the loss of TC4 was about 26 times that of the 0% B_4_C cladding layer, and the loss of TC4 was about 36 times that of the 8% B_4_C cladding layer. Figure 7c shows the micromorphologies of the TC4 substrate and two kinds of cladding layers after friction and wear, respectively.

Due to the low hardness of the TC4 surface, serious plastic deformation occurred during the wear process, and obvious deep furrows were formed along the glide direction. There was an accumulation of abrasive chips at the wear marks, and hard GCr15 abrasive chips were intermingled between the friction pairs, forming three-body abrasive wear, which aggravated the surface wear degree. Heat accumulation generated by friction during the friction process led to adhesion on the friction surface. Under the action of shearing force, the soft material was transferred to the hard material surface, which had the characteristics of abrasive wear and adhesive wear. Compared to TC4, the furrow on the surface of the coating became narrower and shallower, and the wear condition of Ni60 + Cu + Ti (0% B_4_C) was greatly improved. There were wear chips and a small amount of material transfer on the surface of the cladding layer, and the reinforcing phase shed traces during the friction and wear process. Compared with the substrate and the cladding layer without B_4_C added, the fine particles and debris with wear near the furrow had shallow and narrow plough scratches, no material transfer occurred on the surface, and the wear mechanism was micro-abrasive wear. All of the cladding layers could improve the wear resistance of the matrix material, and the cladding layer containing 8% B_4_C had the best protective effect on the matrix material. This is because the hardness of the 8% B_4_C cladding layer was much higher than that of the titanium alloy and had a lower coefficient of friction. The thermal conductivities of TiC and TiB were much higher than that of Ti, which was also one of the factors that improved the wear resistance of the coating. During the wear process, the high hardness of the reinforcing phase could prevent wedging of the wear particle during the friction and wear process and shorten the tear shear point travel of the wear particle on the coating surface. Super elastic NiTi_2_ had good toughness and could transfer the force on the hard-phase particles to other ductile materials in the cladding layer during friction, and the cladding layer released stress through small plastic deformation without fracture, thus significantly reducing the risk of hard-phase particles falling off the cladding layer.

### 3.2. The Effect of CeO_2_ Content

Rare earth elements can improve the fluidity of the molten pool, purify the molten pool, refine the grain, reduce the tendency of cracks in the cladding layer, and improve the overall performance of the cladding layer. In order to obtain a coating with excellent properties, different contents of CeO_2_ were added to the nickel-based composite coating containing 8 wt.% B_4_C, and the effects of CeO_2_ content on the microstructure and properties of the coating were investigated.

#### 3.2.1. Microstructure of the Cladding by CeO_2_ Adding

Figure 8 shows the macroscopic morphologies of cladding layers with different CeO_2_ contents under an optical microscope. It can be seen from the figure that, compared with cladding layers without CeO_2_, the depth of the molten pool was significantly increased and the structure was denser and uniform. The cladding layers with different CeO_2_ contents were well combined with the substrate. The bonding zone of cladding layers with different CeO_2_ contents had no defects such as cracks and pores, indicating that the cladding layer and the substrate formed a strong metallurgical bond. The grain near the fusion line was mainly columnar, and with the decrease in G/R, it mainly formed a fine network structure. Compared to the cladding layer without CeO_2_, the grain was obviously refined and the structure was more uniform. CeO_2_, with a melting point of 2600 °C, is a high melting point compound, which can be used as a nucleating particle during cooling and solidification of the cladding layer and play a role in refining the structure of the cladding layer. The addition of CeO_2_ can enhance the convection of the molten pool. Convection is a type of fluid motion in which heat is transferred by the movement of the fluid itself. Enhanced convection leads to a more uniform distribution of phases in the cladding coating. A homogeneous phase distribution can improve the mechanical properties and performance of the coating. During solidification, rare earth compounds formed due to the presence of CeO_2_ can act as heterogeneous nucleation cores. Nucleation is the process in which solid particles start to form in a liquid. Heterogeneous nucleation cores provide sites for new crystals to form, thus increasing the nucleation rate and leading to a finer microstructure. Nano CeO_2_ particles can inhibit the crystallization and growth of precipitates to a certain extent. This refinement of the microstructure results in coatings with improved microhardness and wear resistance. A finer microstructure means smaller, more uniformly distributed grains, which can contribute to increased hardness and better wear performance [30,31]. The rare earth element Ce accumulated at the grain boundaries as an active point for crystal growth. This accumulation was due to pinning and adsorption effects, which are mechanisms through which particles attach themselves to interfaces or surfaces. The accumulation of Ce at grain boundaries reduced the Gibbs free energy of the entire system. The decrease in Gibbs free energy also reduced the driving force required for grain growth. As a result, grain growth was hindered, which was desirable for obtaining a finer microstructure. By accumulating at grain boundaries, Ce inhibited the diffusion of alloying elements in the molten pool. Inhibiting diffusion hindered crystal growth and intensified the branching of crystals, which again contributed to a finer microstructure [32,33,34].

#### 3.2.2. Microhardness, Friction, and Wear Properties

Figure 9 shows the microhardness distribution curves of laser cladding layers with different CeO_2_ contents from the surface to the base material. It can be observed from the figure that the trend of microhardness distribution in cladding sections with different CeO_2_ contents was basically the same, and the distribution of microhardness values along sections was relatively gentle. The microhardness near the top of the cladding layer was the highest and then slowly decreased, reaching the junction between the cladding layer and the base material, and the hardness value slowly decreased to the hardness of the base material.

With the increase in CeO_2_ content, the thickness of the cladding layer increased from 1.4 mm to 1.9 mm and then decreased to 1.5 mm. When the CeO_2_ content was 3%, the microhardness and average microhardness of the coating were the highest, with the average microhardness being 1105 HV and the highest hardness being 1298 HV. In the laser cladding process, the rare earth oxide CeO_2_ added to the cladding material acted as a grain refiner during the solidification process of the melt pool, improving the nucleation rate and refining the grain structure of the cladding layer, thus significantly improving the comprehensive performance of the cladding layer. In addition, CeO_2_ not only promoted the flow of the molten pool, making the structure of the cladding layer more uniform, but it also improved the absorption rate of laser energy of the cladding material and improved the dilution rate of the cladding layer. An excessive dilution rate will cause the performance of the cladding layer to decline. Therefore, the addition of CeO_2_ to the cladding material is not completely beneficial, but the uniform refinement of the organization, as well as the effect of the increase in TiC and TiB strengthening phases on the hardness and the negative impact of the increase in dilution rate on the hardness compete with each other. When the CeO_2_ content was too low, it could not strengthen the cladding layer. When the CeO_2_ content was 3 wt.%, the advantages of the increase in reinforced phase generation and the grain refinement in the cladding layer outweighed the disadvantages of the increase in dilution rate, so that the microhardness value of the cladding layer was higher than that of the cladding layer without CeO_2_. When the CeO_2_ contents were 4 wt.%, 5 wt.%, and 6 wt.%, the dilution rate was too large and the matrix material melted too much, resulting in dominant negative effects and a reduction in the microhardness of the cladding layer. Especially when the CeO_2_ content was 6 wt.%, the dilution rate of the cladding layer was the highest, and the microhardness value was the lowest due to the greater melting of the matrix material.

When the CeO_2_ content of rare earth oxides was 3%, the microstructure in the cladding layer was the smallest and uniform, the number of reinforcement phases was the largest, and the microhardness of the coating was the highest. Therefore, the cladding layer with 3% CeO_2_ content was selected for tribological and wear performance testing. In order to facilitate comparative analysis, cladding layers without CeO_2_ are also listed in Table 5. Compared to the cladding layer without CeO_2_, the friction coefficient of the cladding layer was slightly decreased, which was 33.7% lower than that of the TC4 matrix material. Therefore, the laser cladding layer with 3% CeO_2_ content of rare earth oxide had the lowest friction coefficient, and the wear reduction performance was relatively optimal. The wear amount of the cladding layer was significantly lower than that of TC4 titanium alloy substrate, and the loss of substrate was about 55 times that of the 3% CeO_2_ cladding. Figure 10 shows the microscopic morphologies of the friction and wear surfaces of the laser cladding layer under scanning electron microscopy when the rare earth oxide CeO_2_ content was 3%. Each reinforcing phase was evenly distributed in the cladding layer, and the furrow on the wear surface was shallow. When the fine hard phase with high hardness in the coating contacted the secondary friction disk, the coating was more difficult to wear down, so the furrow formed was shallow, small, or even terminated, and only a small amount of wear chips existed on the surface. The wear mechanism of the cladding layer was slight abrasive wear.

Further applications of ceramics, intermetallic compounds, or other high hardness materials in TC4 laser cladding are underway. Additionally, composite coatings with multiple functions (such as wear resistance, corrosion resistance, oxidation resistance, etc.) are being further developed.

## 4. Conclusions

In this paper, blown powder laser cladding technology was used to prepare wear-resistant coatings to enhance the surface hardness and wear resistance of TC4 titanium alloy. However, laser cladding technology is prone to cracking in the cladding layer, and single metallic-based coatings no longer meet stringent working conditions. The study optimized the cladding powder by adding Ti, Cu, B_4_C, and CeO_2_ to a Ni60 nickel-based alloy powder. The main conclusions are as follows:

After adding B_4_C, the main phases of the cladding layer consist of NiTi_2_ and Ti_2_Cu metallic compounds and in situ-generated TiC and TiB hard ceramic reinforcing phases. Among them, when the B_4_C content is 8 wt.%, the nickel-based composite coating prepared exhibits the most significant improvement in hardness for the TC4 substrate, with the highest hardness reaching 1377 HV and an average microhardness of 1078 HV, which is 3.17 times that of the TC4 substrate. The average friction coefficient is 0.28, which is 24.7% lower than that of the TC4 substrate. The mass loss of the TC4 substrate is about 36 times that of the cladding layer containing 8 wt.% B_4_C, and the wear mechanism is micro-abrasive wear.

After adding CeO_2_ rare earth oxide, the rare earth element can promote the flow of the molten pool, making the cladding layer more uniform in structure and increasing the thickness of the cladding layer. Extremely fine metallic compounds formed by the Ce element and other elements disperse and strengthen the cladding layer. When the CeO_2_ content is 3 wt.%, the hardness and average hardness of the cladding layer are the highest, with a maximum hardness of 1298 HV and an average microhardness of 1105 HV, which is 3.25 times that of the TC4 substrate. Compared with the substrate material, the average friction coefficient of the 3 wt.% CeO_2_ nickel-based composite coating is 0.25, which is 33.7% lower than that of the TC4 substrate. The weight loss of TC4 is about 55 times that of the 3 wt.% CeO_2_ cladding layer, and the wear mechanism is micro-abrasive wear.

## Figures and Tables

**Figure 1 materials-17-03823-f001:**
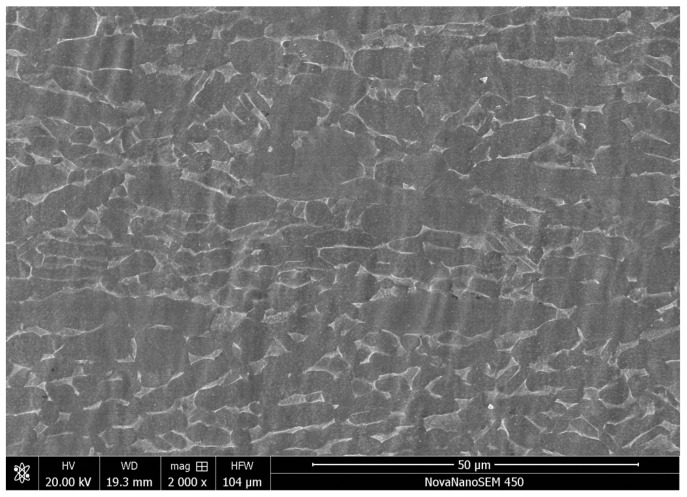
Microstructure of TC4 titanium alloy.

**Figure 2 materials-17-03823-f002:**
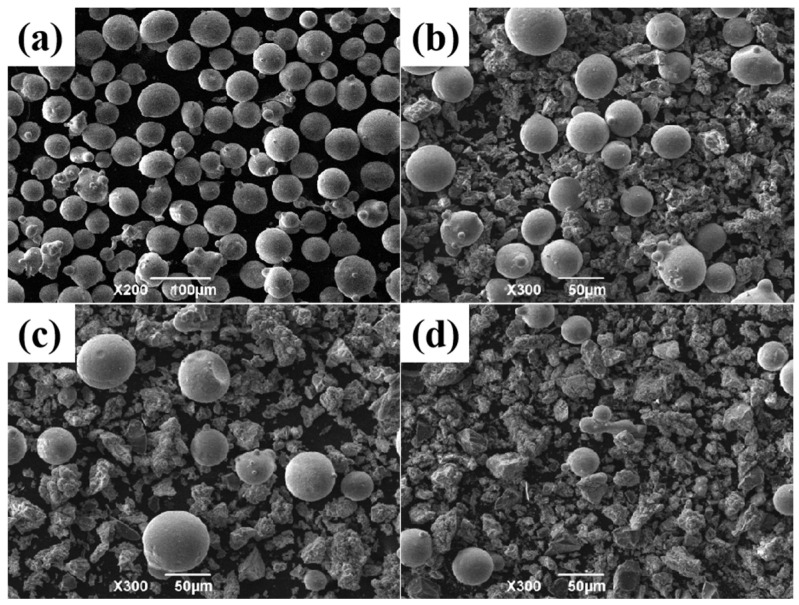
Cladding powders (**a**) Ni60; (**b**) Ni60 + Cu + Ti; (**c**) Ni60 + Cu + Ti + B_4_C; (**d**) Ni60 + Cu + Ti + B_4_C + CeO_2_.

**Figure 3 materials-17-03823-f003:**
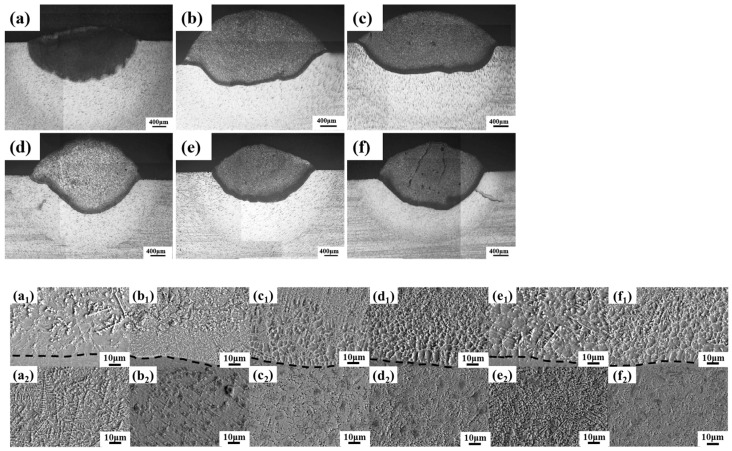
Microstructural morphology of laser cladding layers with varying B_4_C contents: (**a**) 0% B_4_C, (**b**) 2% B_4_C, (**c**) 4% B_4_C, (**d**) 6% B_4_C, (**e**) 8% B_4_C, (**f**) and 10% B_4_C. (**a_1_**–**f_1_**) SEM microstructural mapping near the fusion line, (**a_2_**–**f_2_**) SEM microstructural mapping of the mixing area.

**Figure 4 materials-17-03823-f004:**
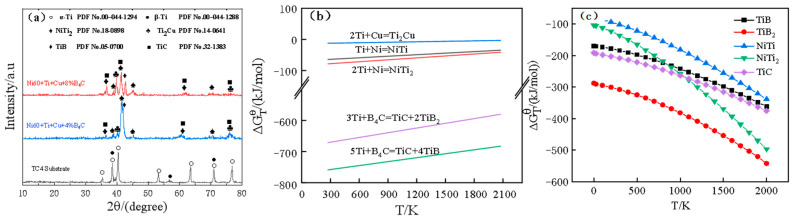
XRD patterns of the cladding layer and variation in Gibbs free energy with temperature. (**a**) X-ray diffraction patterns of the cladding layers (**b**) the standard Gibbs free energy of the reactions (**c**) Gibbs energy change curve of the product at different temperatures.

**Figure 5 materials-17-03823-f005:**
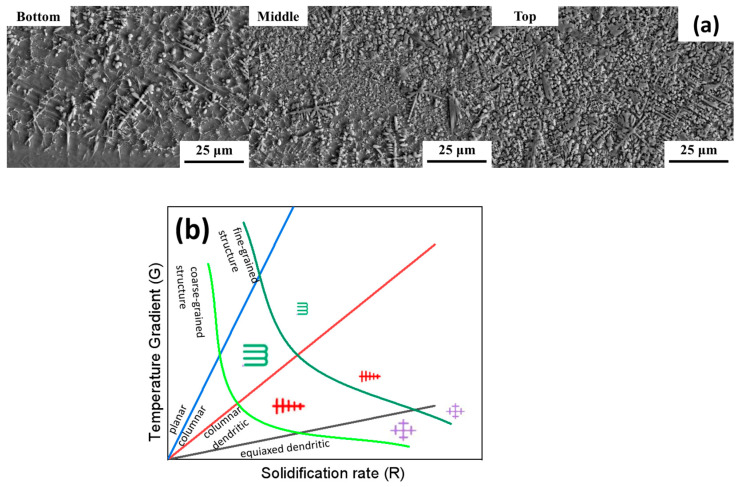
SEM cross-sectional microstructure of 8% B_4_C cladding layer (**a**) and effect of G/R on microstructure (**b**) [22].

**Figure 6 materials-17-03823-f006:**
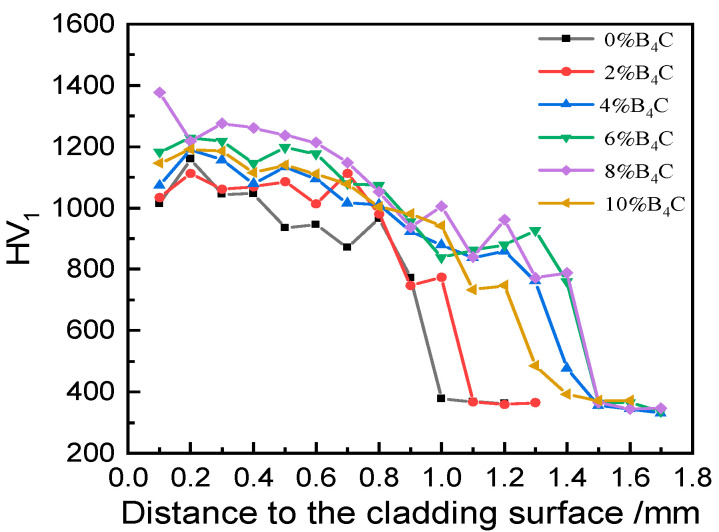
Microhardness distribution of different contents of B_4_C (wt.%).

**Figure 7 materials-17-03823-f007:**
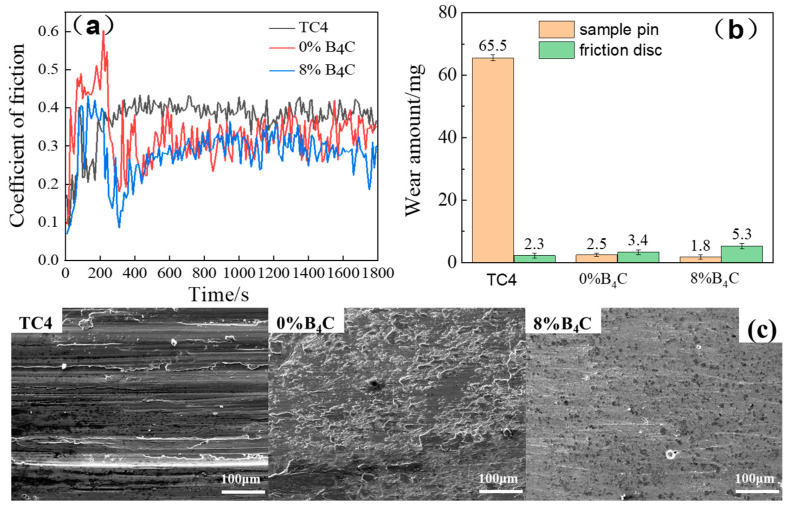
Friction coefficients (**a**), wear amounts (**b**), and micromorphologies (**c**) of the cladding layers.

**Figure 8 materials-17-03823-f008:**
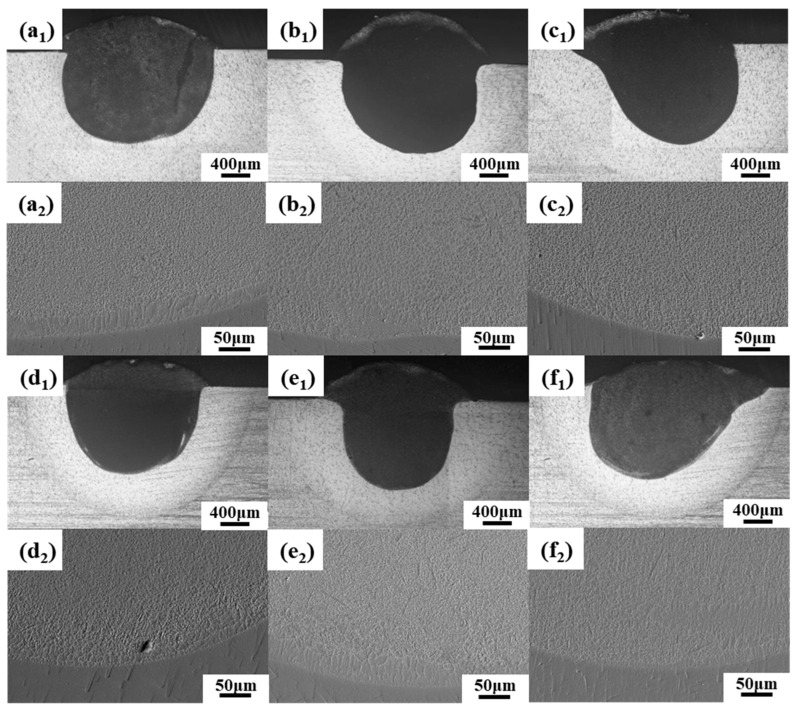
Micromorphologies of cross-sections of cladding layers with different CeO_2_ contents: (**a_1_**,**a_2_**) 1% CeO_2_; (**b_1_**,**b_2_**) 2% CeO_2_; (**c_1_**,**c_2_**) 3% CeO_2_; (**d_1_**,**d_2_**) 4% CeO_2_; (**e_1_**,**e_2_**) 5% CeO_2_; (**f_1_**,**f_2_**) 6% CeO_2_.

**Figure 9 materials-17-03823-f009:**
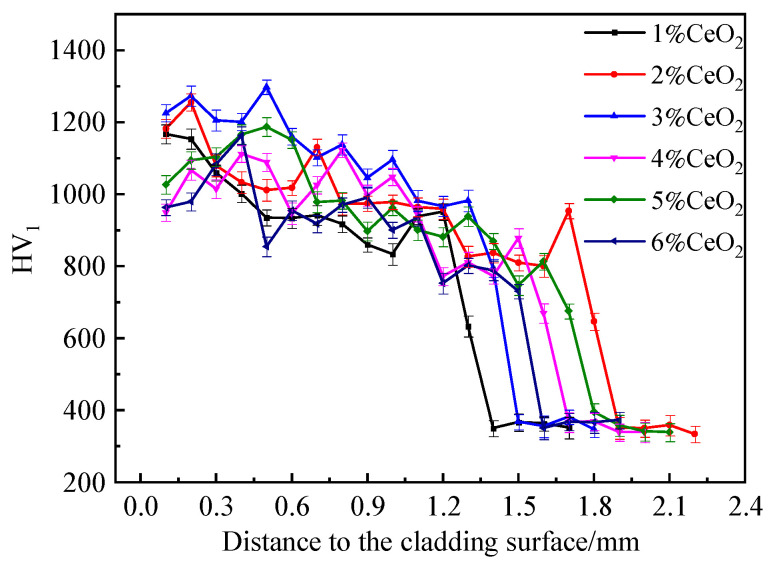
Microhardness distribution of the claddings with different CeO_2_ contents.

**Figure 10 materials-17-03823-f010:**
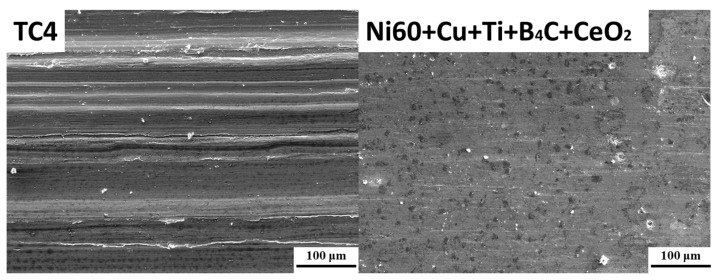
Micromorphologies of the cladding with CeO_2_ addition.

**Table 1 materials-17-03823-t001:** Chemical composition of TC4 titanium alloy (wt.%).

	Al	V	O	Fe	H	C	N	Ti
TC4	6.71	4.23	0.14	0.12	0.03	0.03	0.02	Balance

**Table 2 materials-17-03823-t002:** Chemical composition of Ni60 alloy-based powder (wt.%).

Cr	Fe	Si	B	C	Cu	Ti	Ni
9	2.4	2.1	2.1	0.5	15	25	Balance

**Table 3 materials-17-03823-t003:** Compositions of mixed powders for cladding (wt.%).

No.	Ni60-Ti-Cu	B_4_C
1	100	0
2	98	2
3	96	4
4	94	6
5	92	8
6	90	10

**Table 4 materials-17-03823-t004:** Chemical reactions and products in the Ni-Ti-Cu-B_4_C system.

No.	Chemical Equation of Reaction	Products
1	Ti(s) + Ni(s) = NiTi(s)	NiTi
2	2Ti(s) + Ni(s) = NiTi_2_(s)	NiTi_2_
3	2Ti(s) + Cu(s) = Ti_2_Cu(s)	Ti_2_Cu
4	3Ti(s) + B_4_C(s) = TiC(s) + 2TiB_2_(s)	TiC, TiB_2_
5	5Ti(s) + B_4_C(s) = TiC(s) + 4TiB(s)	TiC, TiB

**Table 5 materials-17-03823-t005:** Friction coefficients and wear amounts of the cladding layers.

Materials	Friction Coefficient	Wear Amount/mg
TC4	0.73	65.5
Ni60 + Cu + Ti	0.33	2.5
Ni60 + Cu + Ti + 8%B_4_C	0.28	1.8
Ni60 + Cu + Ti + 8%B_4_C + 3%CeO_2_	0.25	1.2

## Data Availability

The original contributions presented in the study are included in the article, further inquiries can be directed to the corresponding authors.

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
