# Peer review of "The Influence of Adding B4C and CeO2 on the Mechanical Properties of Laser Cladding Nickel-Based Coatings on the Surface of TC4 Titanium Alloy"

_materials, 2024, doi:10.3390/ma17153823_

Round 1
Reviewer 1 Report
Comments and Suggestions for Authors
The paper is interesting. Authors present results of test made on the wear resistant coating on TC4 substrate with the use of selected particles.
First, TC4 is matrix material or rather substrate. Matrix for coating is Ni60.
All figure must be bigger and with higher resolution In the present for they are completely unreadable.
Scale on SEM images is not visible.
What it mean:
“high purity argon gas is used for protection” or as shielding gas. Is Argon used for powder transportation too?
‘surface needs to be polished with sandpaper” – polished or grinding?
“corrode the metallographic sample for 10 s” – corrode (natural process of material degradation) or etched (process to reveal the microstructure)?
Binding zone or rather fusion line?
Line 185 – “obvious cracks”- What it mean? Cracks are in the HAZ, so it is not the problem of coating properties but insufficient process parameters (in my opinion). Line 192 – how the interatomic diffusion was observed. There are two molten metals, so there is mixing not diffusion.
“absorb heat and melt to form a molten pool for metallurgical chemical reaction” I don’t understand what happened. The main direction for cooling if heat transfer (flow) to the substrate. That why the HAZ is observed during cladding process. Metallurgical chemical reaction – where in molten metal or in solid metal. Maybe precipitation?
What shows Gibbs free energy. Is is true for cladding process with high solidification rate and high cooling rate. This part (line 210-228) should be in introduction or in discussion part. Not next to test results.
Fig. 4 unreadable. Please improve resolution.
Please show the microstructure of claddings. Only the macrostructure is shown.
At the end – please use only passive form.
Author Response
Dear Reviewer,
Thank you very much for taking the time to review my manuscript and for providing valuable feedback. I have carefully considered your comments and have made the necessary revisions to the paper with yellow background. Below are my responses to the main issues you raised along with the corresponding modifications:
- TC4 is matrix material or rather substrate. Matrix for coating is Ni60.
Response: The authours have made detailed revisions to the statement of matrix material and substrate in Line 91, 93, etc.
- All figure must be bigger and with higher resolution In the present for they are completely unreadable.
Response: The authours have adjusted the size and clarity of all images for better readability in Fig1, 2, etc.
- Scale on SEM images is not visible.
Response: The authours have re-added SEM images with the original scale markers in Fig5, 7, etc.
- “high purity argon gas is used for protection” or as shielding gas. Is Argon used for powder transportation too?
Response: High purity argon gas is used as shielding gas and for powder transportation. I have made the corresponding modifications in the manuscript in Line 123.
- ‘surface needs to be polished with sandpaper” – polished or grinding?
Response: The authours have made the corresponding modifications in the manuscript in Line 127.
- “corrode the metallographic sample for 10 s” – corrode (natural process of material degradation) or etched (process to reveal the microstructure)?
Response: The authours have made the corresponding modifications in the manuscript in Line 128, 130.
- Binding zone or rather fusion line?
Response: The authours have made the corresponding modifications in the manuscript in Line 182, 358.
- Line 185–“obvious cracks”- What it mean? Cracks are in the HAZ, so it is not the problem of coating properties but insufficient process parameters (in my opinion). Line 192–how the interatomic diffusion was observed. There are two molten metals, so there is mixing not diffusion.
Response: The occurrence of cracks in the heat-affected zone (HAZ) is related to both the welding material and the welding process. Since the same process parameters are used, the focus here is on the influence of the cladding material. The authours have made the corresponding modifications about “mixing” in Line 192, 233, 240.
- “absorb heat and melt to form a molten pool for metallurgical chemical reaction” I don’t understand what happened. The main direction for cooling if heat transfer (flow) to the substrate. That why the HAZ is observed during cladding process. Metallurgical chemical reaction–where in molten metal or in solid metal. Maybe precipitation?
Response: The authours have made the corresponding modifications in Line 201-205.
- 4 unreadable. Please improve resolution.
Response: The authours have made the corresponding modifications in Fig. 4.
- Please show the microstructure of claddings. Only the macrostructure is shown.
Response: The microstructure of claddings is shown in Fig. 4.
- At the end – please use only passive form.
Response: The authours have made revisions to the manuscript's grammar, with a focus on amending the use of passive voice where appropriate.
I hope that the revisions I have made address your concerns. Should you have any further questions or suggestions, please do not hesitate to let me know, and I will do my best to improve the manuscript. Once again, thank you for your valuable input on my paper.
Sincerely,
Shanna Xu
Reviewer 2 Report
Comments and Suggestions for Authors
The manuscript "The influence of adding B4C and CeO2 on the mechanical properties of laser cladding nickel-based coatings on the surface of TC4 titanium alloy" discusses the enhancement of the surface hardness and wear resistance of TC4 titanium alloy using blown powder laser cladding technology. It investigates the effects of adding Ti, Cu, B4C, and CeO2 to Ni60 nickel-based alloy powder on the properties of the cladding layer. The work presented by the authors is extensive and promising, but before recommending it for publication, several aspects need significant revision, structural reorganization, and the following points need to be addressed:
The introduction should focus more on the main idea of the research, as there are some jumps in the narrative. It is highly recommended to significantly expand the list of cited sources, particularly by including more studies in the field of laser cladding and recent advancements.
- Authors should give more attention in the Introduction and subsequent sections to why certain additives were chosen to improve properties and how these additives function. Additionally, the parameters for laser cladding with various additives should be described and analyzed. How do rare earth oxides like CeO2 promote the flow of the molten pool and contribute to a more uniform cladding layer structure? What are the specific metallic compounds formed by Ce elements that contribute to the dispersion and strengthening of the cladding layer?
- It would be helpful if the authors provide a table summarizing the laser processing parameters, such as speed and power of the laser, as well as other necessary characteristics.
- There is a considerable of technical data that could be moved to SI or condensed to present the material more concisely.
- The scales on Figure 3 are not discernible and need improvement. Ensure that the indices in chemical formulas are consistently formatted as subscript.
- When evaluating the processes occurring during laser cladding with various B4C additives, do the authors consider the temperatures reached at the laser spot focus, and how was this evaluated?
- It is excellent that the authors provide chemical reactions and products in the Ni-Ti-Cu-B4C system in Table 4, but were all these compounds identified using the employed analytical methods? How does the addition of B4C lead to the formation of NiTi2 and Ti2Cu metallic compounds along with in-situ generated TiC and TiB hard ceramic reinforcing phases? It would be interesting to perform mapping and elemental analysis of the sample from the top to the substrate to track the formation of individual phases and inclusions.
- What mechanisms contribute to the significant increase in hardness when 8 wt.% B4C is added to the nickel-based composite coating? How does the addition of B4C affect the microstructure of the cladding layer, particularly in terms of phase distribution and grain size?
In conclusion, while the manuscript is of practical interest, there is a noticeable lack of testing for the synthesized materials and coatings. For example, how do the enhanced properties of the cladding layers translate to real-world applications and performance improvements for titanium alloy components? What are the potential industrial applications that could benefit from the improved hardness and wear resistance provided by these cladding layers? By addressing these points, the paper can be significantly improved and provide a clearer understanding of the research findings and their implications.
Author Response
Dear Reviewer,
Thank you very much for taking the time to review my manuscript and for providing valuable feedback. I have carefully considered your comments and have made the necessary revisions to the paper with green background. Below are my responses to the main issues you raised along with the corresponding modifications:
- The introduction should focus more on the main idea of the research, as there are some jumps in the narrative. It is highly recommended to significantly expand the list of cited sources, particularly by including more studies in the field of laser cladding and recent advancements.
Response: I have made revisions to the introduction section.
- Authors should give more attention in the Introduction and subsequent sections to why certain additives were chosen to improve properties and how these additives function. Additionally, the parameters for laser cladding with various additives should be described and analyzed. How do rare earth oxides like CeO2 promote the flow of the molten pool and contribute to a more uniform cladding layer structure? What are the specific metallic compounds formed by Ce elements that contribute to the dispersion and strengthening of the cladding layer?
Response: The anthours add “The addition of an appropriate amount of CeO2 could improve the convection of the molten pool and promote the uniform distribution of the phases in the cladding coatings. In addition, the rare earth compounds formed during the solidification process could be used as heterogeneous nucleation cores to improve the nucleation rate in the molten pool. Nano-CeO2 could inhibit the crystallization and growth of the precipitates to a certain extent, thereby obtaining a refined microstructure where the coating exhibited improved microhardness and wear resistance. The rare earth element Ce could accumulate as the active point of crystal growth in the grain boundary via pinning and adsorption effects, which reduced the Gibbs free energy of the whole system and the driving force required for grain growth, thereby inhibiting the diffusion of alloying elements in the molten pool to hinder the growth of crystals and intensify their branching” and the relavent citings in the manuscript.
- It would be helpful if the authors provide a table summarizing the laser processing parameters, such as speed and power of the laser, as well as other necessary characteristics.
Response: The authors have already published an article in the Xinjiang Nonferrous Metals Journal regarding the optimization of laser processing parameters. DOI:10.16206/j.cnki.65-1136/tg.2024.02.037.
- There is a considerable of technical data that could be moved to SI or condensed to present the material more concisely.
Response:
- The scales on Figure 3 are not discernible and need improvement. Ensure that the indices in chemical formulas are consistently formatted as subscript.
Response: The authours have made the corresponding modifications in Fig. 3.
- When evaluating the processes occurring during laser cladding with various B4C additives, do the authors consider the temperatures reached at the laser spot focus, and how was this evaluated?
Response: The authors did not measure the temperature at the laser focal point in real-time during the experiment, which is also an aspect to be mindful of in subsequent work. In the experiments with different B4C additions, the same process parameters were used, so it is believed that the heat input was consistent.
- It is excellent that the authors provide chemical reactions and products in the Ni-Ti-Cu-B4C system in Table 4, but were all these compounds identified using the employed analytical methods? How does the addition of B4C lead to the formation of NiTi2 and Ti2Cu metallic compounds along with in-situ generated TiC and TiB hard ceramic reinforcing phases? It would be interesting to perform mapping and elemental analysis of the sample from the top to the substrate to track the formation of individual phases and inclusions.
Response: Thank you for your suggestion. We will conduct further research on the phase change from the top to the bottom of the cladding layer in our subsequent work.
- What mechanisms contribute to the significant increase in hardness when 8 wt.% B4C is added to the nickel-based composite coating? How does the addition of B4C affect the microstructure of the cladding layer, particularly in terms of phase distribution and grain size?
Response: TiC (titanium carbide) and TiB (titanium boride) ceramic phases can significantly increase the hardness of nickel-based cladding coatings. The mechanisms of these ceramic phases' effect on hardness increase mainly include the following two points: TiC and TiB ceramic phases act as hard particles dispersed in the nickel matrix, effectively hindering the movement of dislocations. When dislocations encounter these hard particles, they tend to bypass the particles rather than cutting through them, thereby increasing the yield strength and hardness of the material; the ceramic phases can serve as nucleation cores, promoting heterogeneous nucleation of the nickel matrix during solidification, thus refining the grain size and enhancing the hardness.
I hope that the revisions I have made address your concerns. Should you have any further questions or suggestions, please do not hesitate to let me know, and I will do my best to improve the manuscript. Once again, thank you for your valuable input on my paper.
Sincerely,
Shanna Xu
Round 2
Reviewer 1 Report
Comments and Suggestions for Authors
Thank you very much for the corrections and answers provided.
Fig. 3 was well labeled in the original text, these are macroscopic photos. I asked to add additional microscopic photos, i.e. at a higher magnification, e.g. from the mixing area or fusion line.
The scale in the SEM photos has not been corrected. Please remove the bar generated by the microscope software and replace it with a scale similar to the one in the photos in Fig. 3.
Scale in Fig. 8 is unreadable. Please make the photos bigger and with higher resolution.
Author Response
Dear Reviewer,
Thank you very much for taking the time to review my manuscript and for providing valuable feedback. I have carefully considered your comments and have made the necessary revisions to the paper with purple-pink background. Below are my responses to the main issues you raised along with the corresponding modifications:
Fig. 3 was well labeled in the original text, these are macroscopic photos. I asked to add additional microscopic photos, i.e. at a higher magnification, e.g. from the mixing area or fusion line.
Response: The authours have added additional microscopic photos from the mixing area and the fusion line area in Fig. 3.
The scale in the SEM photos has not been corrected. Please remove the bar generated by the microscope software and replace it with a scale similar to the one in the photos in Fig. 3.
Response: The authours have replaced the scales in Fig. 5, Fig. 7 and Fig. 10.
Scale in Fig. 8 is unreadable. Please make the photos bigger and with higher resolution.
Response: The authours have adjusted the size and scales in Fig. 8.
I hope that the revisions the authours made address your concerns. Should you have any further questions or suggestions, please do not hesitate to let me know, and I will do my best to improve the manuscript. Once again, thank you for your valuable input on my paper.
Sincerely,
Shanna Xu